# Identification and Characterization of microRNAs in the Gonads of *Litopenaeus vannamei* Using High-Throughput Sequencing

**Wei Li †, Pingping He †, Xingzhi Zhang , Junliang Guan, Yongxian Chen, Li Zhang, Bin Zhang, Yusi Zheng, Xin Li, Qingsong He, Longcheng Liu, Chang Yuan, Pinyuan Wei \* and Jinxia Peng \***

Guangxi Key Laboratory of Aquatic Genetic Breeding and Healthy Aquaculture,
Guangxi Academy of Fisheries Sciences, Nanning 530021, China
\* Correspondence: weipinyuan@hotmail.com (P.W.); pengjinxia@gmail.com (J.P.)
† These authors contributed equally to this work.

**Abstract:** Although the expression of miRNAs has been widely applied to investigate on gonads, the role of miRNAs in the gonadal development of white Pacific shrimp (*Litopenaeus vannamei*) remains unknown. In this study, we performed high-throughput sequencing to identify the sex-related microRNAs (miRNAs) that elucidated the regulatory mechanisms on the gonadal differentiation of *L. vannamei*. We obtained a total of 29,671,557 and 28,526,942 raw reads from the ovaries and testes library, respectively. We then mapped 26,365,828 (92.73%) of the ovarian clean sequences and 23,694,294 (85.65%) of the testicular clean sequences for a transcriptome reference sequence of *L. vannamei*. After blasting the miRNA sequences against the miRBase database, we identified 153 significantly differentially expressed miRNAs between the ovaries and testes. To confirm the high-throughput sequencing results, we used a reverse transcriptase–quantitative polymerase chain reaction (RT-qPCR) to verify the expression patterns of the seven most differentially expressed miRNAs (i.e., novel_mir23, miR-92b-3p_3, miR-12-5p_2, novel_mir67, miR-279_1, let-7-5p_6, miR-263a-5p_1). According to the results of RT-qPCR, most of the miRNAs were expressed consistently with the high-throughput sequencing results. In addition, the target genes significantly enriched several Kyoto Encyclopedia of Genes and Genome (KEGG) pathways that were closely related to gonadal differentiation and development, including extracellular matrix–receptor interaction, Hedgehog signaling pathway, protein digestion and absorption and cell adhesion molecules (CAMs). This study revealed the first miRNAs sequencing of *L. vannamei* gonads. We identified sex-related differentially expressed miRNAs and KEGG pathways, which will be helpful to facilitate future research into the regulatory mechanism on the gonadal differentiation of *L. vannamei*.

**Keywords:** *Litopenaeus vannamei*; miRNAs; gonad; high-throughput sequencing; RT-qPCR

## 1. Introduction

The white Pacific shrimp (*Litopenaeus vannamei*), a tropical shrimp and euryhaline species, originated in the Pacific east coast, spread from northern Mexico, while passing through Central and South America, and extended as far as southernmost Peru (http://www.fao.org/fishery/en, accessed on 8 June 2011). In recent years, *L. vannamei* aquaculture has developed rapidly. It has become one of the three most-widely cultivated shrimp species globally, because of its low demand for nutrition, fast growth, adaptability to a comparatively broad range of salinity, long survival time out of water, and good disease resistance [1]. In 2019, the global production of *L. vannamei* approached about 5,446,216 tons, which increased by 80.2% compared with 2010, according to the Food and Agriculture Organization of the United Nations (FAO data). With the expansion of farming scale, the demand for high-quality broodstocks is also increasing, although supply is limited. To date, many genes that are differentially expressed between ovaries and testes played a critical

role in the regulation of gonadal development. Research about *L. vannamei* has centered primarily on immunologic defense [2,3], disease [4,5], growth [6,7], and culturing techniques [8,9], but the research on its reproductive activity and molecular aspects of gonadal development remains lacking. The decisive factors of sexual development are determined by genetics or environment, or by the coregulation of genetics and environment [10–12]. Gonads, including ovaries and testes, are indispensable reproductive organs. Thus, a comprehensive understanding of the regulatory mechanisms driving sexual differentiation in *L. vannamei* is urgently needed, including miRNAs involved in the gonadal development.

MiRNAs, 20–24 nucleotide (nt) long, small non-coding RNAs, are major participators in epigenetics. They are responsible for post-transcriptionally regulating the expression of genes. High-throughput sequencing technology is generally used to identify and screen miRNAs in diverse species. Studies have suggested that miRNAs play crucial roles in the regulation of growth, reproduction, immunity, and the stress response [13–16]. In organisms, one miRNA with a different expression profile can potentially regulate multiple gene expressions, or can control a single gene expression by multiple miRNAs, depending on the cells and developmental stage [17]. Thus, we inferred that miRNAs are differentially expressed in the ovaries and testes.

In recent years, intensive studies have investigated miRNA expression in the gonads of several different species, including freshwater mussel (*Hyriopsis cumingii*) [18], yellowfin seabream (*Acanthopagrus Latus*) [19], Hong Kong oyster (*Crassostrea hongkongensis*) [20], ovate pompano (*Trachinotus ovatus*) [21], medaka (*Oryzias latipes*) [22], common carp (*Cyprinus carpio*) [23], Black Tiger Shrimp (*Penaeus monodon*) [24], and giant freshwater prawn (*Macrobrachium rosenbergii*) [25]. These studies have revealed that tissue-specific miRNAs are essential for sexual differentiation and development.

No information is available on miRNAs associated with *L. vannamei* gonad differentiation. We selected differentially expressed miRNAs between ovaries and testes using high-throughput sequencing. These data will help further our understanding of the regulation mechanism of gonad differentiation in the *L. vannamei*. In addition, we used a reverse transcription–quantitative polymerase chain reaction (RT-qPCR) to verify differentially expressed miRNAs and to validate the reliability of the high-throughput sequencing.

## 2. Materials and Methods

### 2.1. Experimental Shrimp and Sample Preparation

The 35 shrimps used in this study, including the 19 male shrimps with mean body weight of $46.1 \pm 2.06$ g and mean body length of $14.90 \pm 0.35$ cm, and 16 female shrimps with mean body weight of $62.2 \pm 3.13$ g and mean body length of $16.12 \pm 0.43$ cm, were sourced from *Litopenaeus vannamei* shrimp-breeding farm of Guangxi Academy of Fisheries Sciences. The experimental shrimps were frozen and paralyzed on ice for 3 min, and then their gonads were removed. We immediately soaked a portion of the separated sample in RNAlater and stored the sample at $-80\,^{\circ}$C. Using Trizol reagent (Invitrogen, Carlsbad, CA, USA), we extracted total RNA from these samples. The spare gonads of shrimps were stored in 4% paraformaldehyde for 48 h, and then were dehydrated completely with a gradient of 70–100% ethanol, embedded in paraffin, and serially sectioned (5–10 μm) for hematoxylin and eosin (H&E) staining. We observed the histological sections of shrimps using light microscopy to verify the sex and gonadal development stage.

### 2.2. Small RNA cDNA Library Construction by High-Throughput Sequencing

Firstly, we mixed ovarian and testicular RNA samples equally (three RNA samples from the ovary and testis group, respectively) and generated a pooled female and male RNA sample. Then, we used 3 μg of RNA per pooled samples to construct two small RNA libraries using Small RNA Sample Pre Kit (BGI Gene, Shenzhen, China). Then, the sRNA were separated from gonadal total RNA, and these sRNA were subsequently ligated to 5′ and 3′ adaptors using the reverse transcription (RT) primer with unique molecular identifier. With RT, we used products to synthesize one-strand cDNA, which were amplified by highly

sensitive PCR. Using PCR products (gel percentages 2.5%), we separated approximately 110–130 bp by polyacrylamide gels electrophoresis. Then, we conducted library quantitative and pooling cyclization for the fragment selected. We evaluated the library quality by DNA Nano Ball (DNB), and then sequenced the library on the BGISEQ-500 platform.

### 2.3. Bioinformatics Analysis

After removing low-quality tags, tags with 5′ primer contaminants and poly A, tags without 3′ primer and insertion, we reserved tags of 18–30 nt for the following analysis. We mapped these clean tags to the *L. vannamei* transcriptome reference sequence (https://www.ncbi.nlm.nih.gov/nuccore/GDUV00000000.1/, accessed on 5 October 2016) and other small RNA databases by AASRA [26]. Then, we compared the mapped small RNA sequences against the Rfam database, to remove tRNAs, rRNAs, snRNA, noRNAs and repeat sequences. Then, we searched the remaining small RNA sequences in miRBase to identify the known miRNAs and to predict the novel miRNAs using Mirdeep2 [27]. We normalized the miRNA expression levels by transcripts per million (TPM). We predicted the target genes of miRNAs by RNAhybrid [28], miRanda [29] and TargetScan [30]. We identified differential expression analysis of miRNAs by the DESeq. Adjusted *p*-values < 0.05 and the absolute value of Log2Ratio $\geq 0$ were set as the standard for significantly differential expression. We used miRanda to predict the target genes of miRNAs [29]. Gene ontology (GO) annotation and Kyoto Encyclopedia of Genes and Genomes (KEGG) pathway were used to analyze the target gene enrichment [31].

### 2.4. Stem-Loop RT-qPCR

To validate differentially expressed miRNAs identified from sequencing data between the ovaries and testes of *L. vannamei*, we used stem-loop RT-qPCR to quantify the relative expression levels of the seven miRNAs (Table 1). RNA sample, RT-qPCR reagents (Takara, R820A), and the specific primers listed in Table 2 were used to perform RT-qPCR on the ABI 7500 PCR system (Applied Biosystems, Foster City, CA, USA). The RT-qPCR program was as follows: predenaturation for 20 s at 95 °C, denaturation at 95 °C for 15 s and extension at 55 °C for 30 s for 40 cycles. All samples were biologically replicated in triplicate, and miRNAs' relative expression levels were calculated with the $2^{-\Delta\Delta CT}$ strategy. We set U6 as an internal control. We used SPSS version 16.0 (IBM, CHI, USA) to calculate difference by one-way analysis of variance (ANOVA). Before using ANOVA, data were tested for normal distribution, and assumed homogeneity of variances. It is observed that data is homogeneous and has a normal distribution. After the earlier analysis, the descriptive statistics and one-way variance analysis were applied in data analysis. A value of *p* < 0.05 was considered statistically significant.

**Table 1.** The differentially expressed miRNAs of *L. vannamei* identified by high-throughput sequencing and verified by RT-qPCR.

| miRNA id | Expression (Testis) | Expression (Ovary) | log$_2$FoldChange (Ovary/Testis) |
|---|---|---|---|
| novel_mir23 | 6437.118799 | 0.508025321 | −13.62922707 |
| miR-92b-3p_3 | 22,983.4716 | 1102.313603 | −4.381990104 |
| miR-12-5p_2 | 969.0430681 | 217.7167985 | −2.154108059 |
| novel_mir67 | 0.172733727 | 1287.990965 | 12.86428505 |
| miR-279_1 | 418.747114 | 4246.721485 | 3.342198342 |
| let-7-5p_6 | 414.5091587 | 28,964.08332 | 6.126717217 |
| miR-263a-5p_1 | 5.9993158 | 156.8735167 | 4.708660031 |

**Table 2.** The primer sequences of differentially expressed miRNAs of *L. vannamei* used for RT-qPCR.

| MiRNAs | Mature Sequence (5′-3′) | | Primer Sequences (5′-3′) |
|---|---|---|---|
| novel_mir23 | GAGGACGUGGUAG CCUAGUGGU | Fwd Rev Stem | GCGGAGGACGTGGTAGCCT AGTGCAGGGTCCGAGGTATT GTCGTATCCAGTGCAGGGTC CGAGGTATTCGCACTGGATACGACACCACT |
| miR-92b-3p_3 | AATTGCACTAGTC CCGGCCTGC | Fwd Rev Stem | GCGCGAAGCACAGCCC AGTGCAGGGTCCGAGGTATT GTCGTATCCAGTGCAGG GTCCGAGGTATTCGCACTGGATACGACGCGGCC |
| miR-12-5p_2 | TGAGTATTACATC AGGTACTGGT | Fwd Rev Stem | GCGCGCGGAGAACACA AGTGCAGGGTCCGAGGTATT GTCGTATCCAGTGCAGG GTCCGAGGTATTCGCACTGGATACGACCCGTCC |
| novel_mir67 | UGAUGAGGUCUU GUCGUGAGGAGU | Fwd Rev Stem | GCGTGATGAGGTCTTGTCGTG AGTGCAGGGTCCGAGGTATT GTCGTATCCAGTGCAGGGTCC GAGGTATTCGCACTGGATACGACACTCCT |
| miR-279_1 | TGACTAGATCCAC ACTCATCCA | Fwd Rev Stem | CGCGCGGACAGACCACA AGTGCAGGGTCCGAGGTATT GTCGTATCCAGTGCAGGG TCCGAGGTATTCGCACTGGATACGACTGGTGG |
| let-7-5p_6 | TGAGGTAGTAGGT TGTATAGTT | Fwd Rev Stem | GCGCGCGCGGAGGAGA AGTGCAGGGTCCGAGGTATT GTCGTATCCAGTGCAGG GTCCGAGGTATTCGCACTGGATACGACCTTCCC |
| miR-263a-5p_1 | AATGGCACTGGAA GAATTCACGG | Fwd Rev Stem | CGCGAAGGCACGGAAGA AGTGCAGGGTCCGAGGTATT GTCGTATCCAGTGCAGGG TCCGAGGTATTCGCACTGGATACGACCCGTGT |
| U6 | | Fwd Rev | CTCGCTTCGGCAGCACA AACGCTTCACGAATTTGCGT |

## 3. Results

### 3.1. Determination of L. vannamei Sex and Development Stages Based on Gonad Sections

The gonad section results verified that of the six shrimps used for high-throughput sequencing, three were female and three were male. Of these, three males were at stage III (Figure 1C), with testes mainly containing spermatocytes and spermatids cell; and three females were at stage III with ovaries primarily containing vitellogenetic oocyte (Figure 1D). Six female and six male shrimps were used for RT-qPCR. Among these, three males and three females were at stage I (Figure 1A,B), with testes and ovaries mostly including spermatogonia, oogonia, and oocytes before vitellogenesis, and three males and three females at stage III [32–34].

### 3.2. Summary of Sequencing Data in the Gonads of L. vannamei

We constructed cDNA libraries for sequencing from samples in shrimps. In total, we generated 29,671,557 and 28,526,942 raw reads from the ovaries and testes library, respectively. After removing low-quality tags, tags with 5′ primer contaminants and poly A, tags without 3′ primer and insertion, we had a remaining 28,431,612 and 27,665,604 clean reads that were 17–32 nt long, from the ovaries and testes library, respectively. In the ovaries, clean reads were peaked at 27 nt, followed by 26 nt and 28 nt, whereas in testes, the peak was at 22 nt, followed by 26 nt and 27 nt (Figure 2A). Of these clean reads, 26,365,828 (92.73%) of the ovarian clean sequences and 23,694,294 (85.65%) of the testicular clean

sequences were mapped to the transcriptome reference sequence of *L. vannamei* (GenBank accession number GDUV00000000).

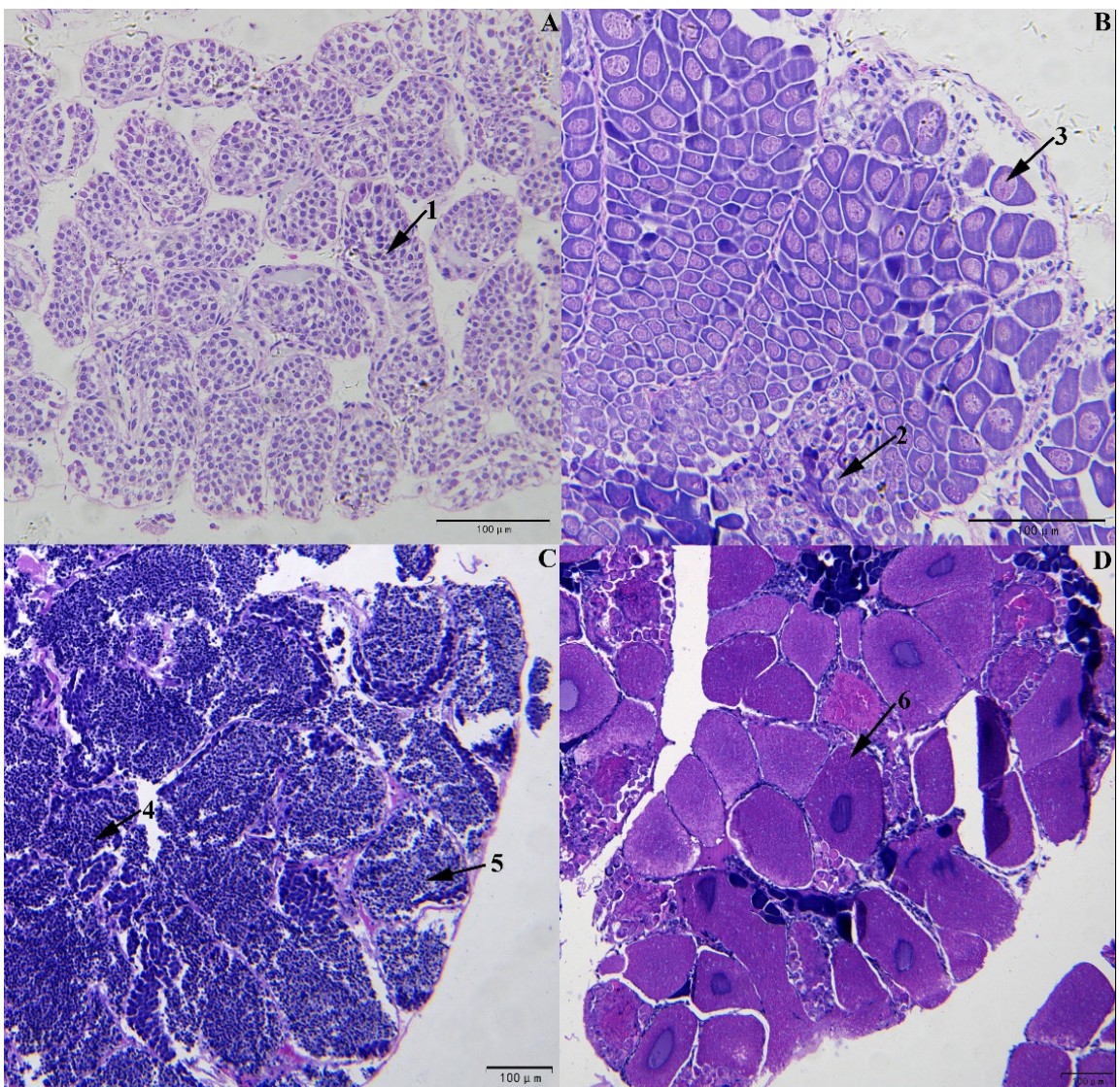

**Figure 1.** Sections of gonads in *L. vannamei* at different stages. (**A**). Testis at stage I. (**B**). Ovary at stage I. (**C**). Testis at stage III. (**D**). Ovary at stage III. Arrows: 1. spermatogonia; 2. oogonia; 3. oocytes before vitellogenesis; 4. spermatocytes; 5. spermatids; 6. vitellogenetic oocyte.

### 3.3. Identification and Prediction of miRNAs in the L. vannamei Ovaries and Testes

To verify the known miRNAs in the gonads of *L. vannamei*, we compared the mapped sequences in the miRBase database. Across the ovaries and testes, we identified 106 and 117 known mature miRNAs, respectively, for the ovaries and testes libraries. According to the mirdeep2 prediction result, we predicted 159 and 214 novel miRNAs from the ovaries and testes libraries, respectively. These miRNAs in the gonadal libraries indicated abundant expression levels. Of these, several miRNAs (i.e., miR-100_2, let-7-5p_3, miR-279_1 and novel_mir71) were highly expressed with thousands of reads in both ovaries and testes. Some miRNAs (i.e., miR-252a-3p_2, miR-210a_1, miR-281-3p and novel_mir133), however, demonstrated only a dozen reads across the two libraries.

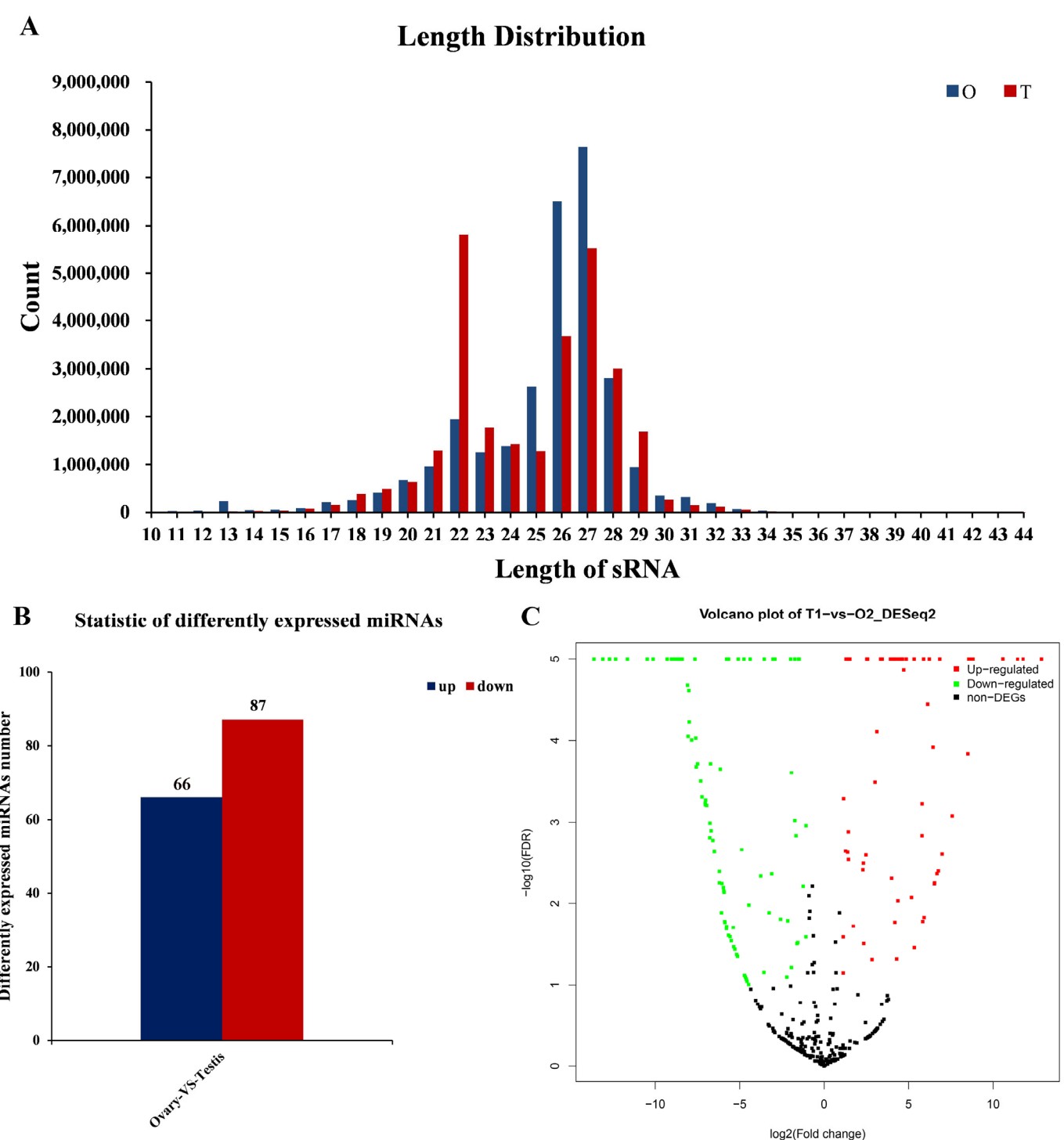

**Figure 2.** Identification of miRNAs. (**A**) Length distribution of clean reads of *L. vannamei*, the letter O represents ovary, and the letter T represents testis. (**B**) Statistic of differently expressed miRNAs in gonads of *L. vannamei*; the blue bar represents the number of up-regulated miRNAs in ovaries, and the red bar represents the number of down-regulated miRNAs in ovaries. (**C**) Volcano plot of differentially expressed miRNAs in ovary and testis of *L. vannamei*.

### 3.4. Screening Differentially Expressed miRNAs in the Gonads of L. vannamei

A total of 153 miRNAs were significantly differentially expressed between ovaries and testes ($|\log2$ (foldchange)$| > 0$, Padj $< 0.05$). In total, 66 significantly up-regulated miRNAs and 87 significantly down-regulated miRNAs were identified in ovaries (Figure 2B,C). Of these, the expression of novel_mir23 in the testes was more than 12,670-fold higher than the ovaries, while the expression of novel_mir128 in ovaries was more than 3493-fold higher than in testes. All of the differentially expressed miRNAs are listed in Table S1.

To confirm high-throughput sequencing results, we used RT-qPCR to verify the expression patterns of the seven most differentially expressed miRNAs (i.e., novel_mir23, miR-92b-3p_3, miR-12-5p_2, novel_mir67, miR-279_1, let-7-5p_6, miR-263a-5p_1). According RT-qPCR results, most of the miRNAs were expressed consistently with the high-throughput sequencing data. Overall, the RT-qPCR results validated those of the high-throughput sequencing and offered support for the reliability of the differentially expressed miRNAs.

These miRNAs exhibited sex-specific expression between ovaries and testes. The miR-NAs let-7-5p_6 ($F_{3,11} = 275.5$, $p < 0.001$), novel_mir67 ($F_{3,11} = 197.1$, $p < 0.001$), miR-12-5p_2 ($F_{3,11} = 282.7$, $p < 0.001$), miR-279_1 ($F_{3,11} = 58.52$, $p < 0.001$), and miR-92b-3p_3 ($F_{3,11} = 3057$, $p < 0.001$) were significantly up-regulated in ovaries, and more highly expressed in ovaries at stage I than stage III (Table 3, Figure 3). The novel_mir23 ($F_{3,11} = 280.5$, $p < 0.001$) were up-regulated in testes; however, its expression level was higher in testes at stage I than stage III (Table 3, Figure 3). The miR-263a-5p_1 ($F_{3,11} = 36.11$, $p < 0.001$) was highly expressed in both ovaries and testes(Table 3, Figure 3). Interestingly, in gonadal stage I, it was more highly expressed in testes than in ovaries, while in gonadal stage III, it was more highly expressed in ovaries than in testes (Figure 3).

**Table 3.** Analysis of variance of relative expression levels of the seven miRNAs confirmed using RT-qPCR.

| miRNA_id | Source of Variance | Sum of Squares | Degrees of Freedom | Mean Square | F Value | *p*-Value |
|---|---|---|---|---|---|---|
| let-7-5p_6 | Regression | 11.54 | 3 | 3.846 | 275.5 | 0.000 |
| | Residual | 0.1117 | 8 | 0.01396 | | |
| | Total | 11.65 | 11 | | | |
| miR-263a-5p_1 | Regression | 0.3094 | 3 | 0.1031 | 36.11 | 0.000 |
| | Residual | 0.02285 | 8 | 0.002856 | | |
| | Total | 0.3322 | 11 | | | |
| miR-279_1 | Regression | 0.8922 | 3 | 0.2974 | 58.52 | 0.000 |
| | Residual | 0.04065 | 8 | 0.005082 | | |
| | Total | 0.9328 | 11 | | | |
| novel_mir67 | Regression | 4.534 | 3 | 1.511 | 197.1 | 0.000 |
| | Residual | 0.06134 | 8 | 0.007667 | | |
| | Total | 4.596 | 11 | | | |
| miR-12-5p_2 | Regression | 6.314 | 3 | 2.105 | 282.7 | 0.000 |
| | Residual | 0.05956 | 8 | 0.007445 | | |
| | Total | 6.374 | 11 | | | |
| miR-92b-3p_3 | Regression | 63.06 | 3 | 21.02 | 3057 | 0.000 |
| | Residual | 0.05501 | 8 | 0.006877 | | |
| | Total | 63.11 | 11 | | | |
| novel_mir23 | Regression | 26.51 | 3 | 8.837 | 280.5 | 0.000 |
| | Residual | 0.2521 | 8 | 0.03151 | | |
| | Total | 26.76 | 11 | | | |

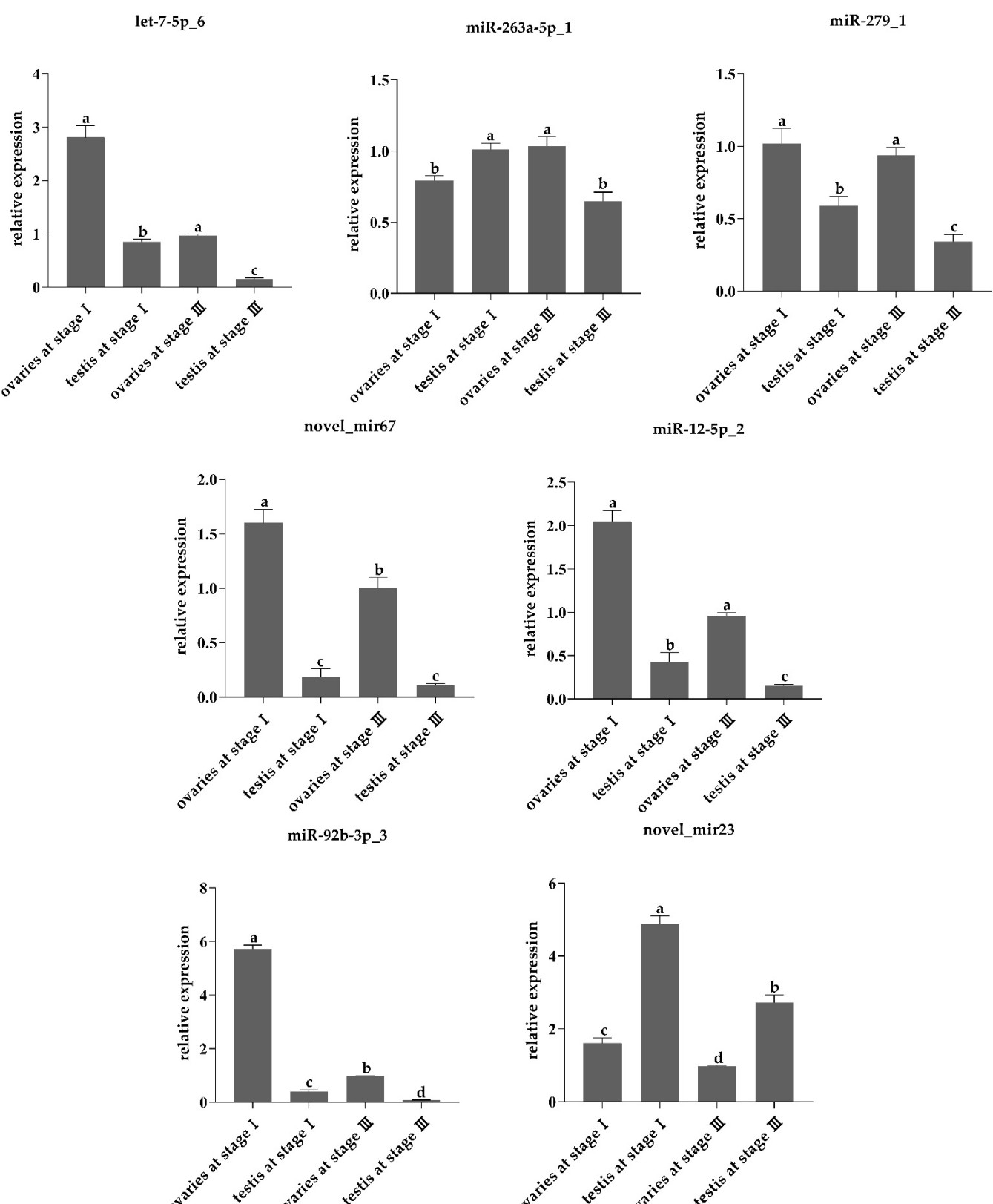

**Figure 3.** The relative expression of differentially expressed miRNAs at different gonadal stage of *L. vannamei*. The expression was calculated through comparative CT (ΔΔCT) methods utilizing *U6* as a reference. The bars showing means +/− standard error. Different letters above the bars indicated significant difference ($p < 0.05$).

*3.5. The Differentially Expressed miRNAs Targets Prediction and Functional Annotation*

To demonstrate the mechanism and biological function of differentially expressed miRNAs in the ovaries and testes of *L. vannamei*, we predicted the target genes using miRanda [29], and also performed functional annotation as well. We predicted a total of 57,412 target genes from all 153 differentially expressed miRNAs obtained. The enrichment degree of functional genes was determined by GO annotation analysis and KEGG pathway analysis. The GO annotations included biological process (BP), cellular components, and molecular function (MF). At the BP level, the target genes were enriched in the GO terms "cellular process" and "metabolic process"; at the CC level, the target genes were enriched in the GO terms "membrane", "membrane part" and "cell"; at the MF level, the target genes were enriched in the GO terms "binding", and "catalytic activity" (Figure 4).

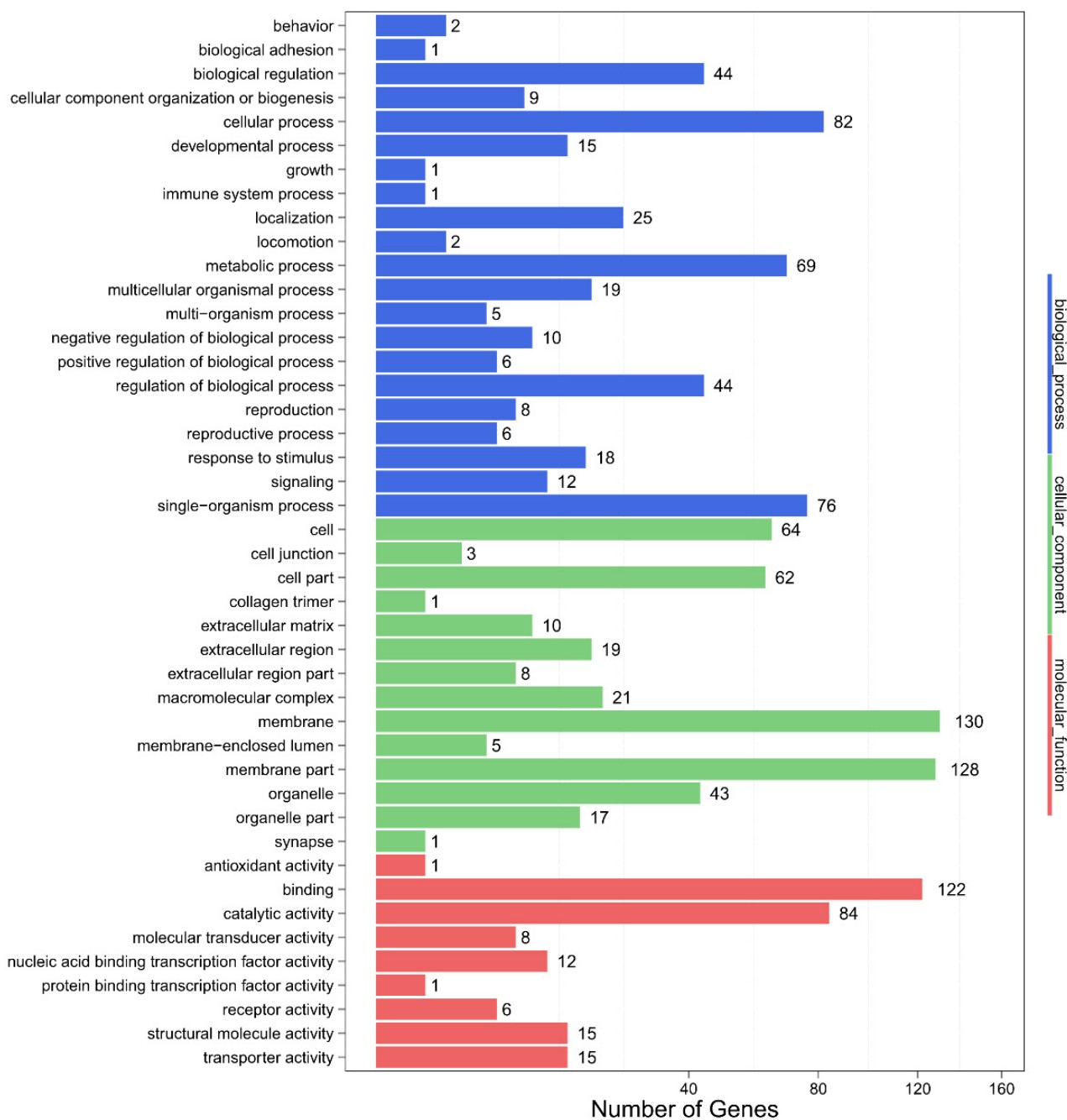

**Figure 4.** GO annotation of target genes of the differentially expressed miRNAs of *L. vannamei*.

The KEGG pathway analysis showed that the target genes were significantly enriched in 343 pathways. The results revealed that the top 20 KEGG pathways included IL-17 signaling pathway, Hepatitis B, extracellular matrix (ECM)-receptor interaction, Hedgehog signaling pathway, prostate cancer, protein digestion and absorption, and cell adhesion molecules (CAMs) (Table 4).

**Table 4.** The top 20 KEGG pathways enriched (*q*-value $\leq$ 0.5) by target genes of differentially expressed miRNAs of *L. vannamei*.

| Pathway-Term | *p*-Value | *q*-Value | Pathway ID |
|---|---|---|---|
| IL-17 signaling pathway | $8.68 \times 10^{-16}$ | $2.98 \times 10^{-13}$ | ko04657 |
| Hepatitis B | $3.42 \times 10^{-7}$ | $5.87 \times 10^{-5}$ | ko05161 |
| Endocrine resistance | $8.08 \times 10^{-7}$ | $9.24 \times 10^{-5}$ | ko01522 |
| Relaxin signaling pathway | $3.666 \times 10^{-6}$ | 0.0003144 | ko04926 |
| AGE-RAGE signaling pathway in diabetic complications | $4.594 \times 10^{-6}$ | 0.0003152 | ko04933 |
| Parathyroid hormone synthesis, secretion and action | $6.455 \times 10^{-6}$ | 0.000369 | ko04928 |
| Other glycan degradation | $1.177 \times 10^{-5}$ | 0.0005062 | ko00511 |
| Phospholipase D signaling pathway | $1.181 \times 10^{-5}$ | 0.0005062 | ko04072 |
| MicroRNAs in cancer | $1.905 \times 10^{-5}$ | 0.0007261 | ko05206 |
| Insulin resistance | $2.48 \times 10^{-5}$ | 0.0008508 | ko04931 |
| ECM-receptor interaction | $3.992 \times 10^{-5}$ | 0.0012447 | ko04512 |
| Human papillomavirus infection | 0.0001299 | 0.0034973 | ko05165 |
| Amoebiasis | 0.0001325 | 0.0034973 | ko05146 |
| Hedgehog signaling pathway | 0.0003282 | 0.0080415 | ko04340 |
| Glycosaminoglycan biosynthesis—keratan sulfate | 0.0006459 | 0.0147702 | ko00533 |
| Prostate cancer | 0.0011405 | 0.0241681 | ko05215 |
| Protein digestion and absorption | 0.0012572 | 0.0241681 | ko04974 |
| Cell adhesion molecules (CAMs) | 0.0012683 | 0.0241681 | ko04514 |
| Insect hormone biosynthesis | 0.001342 | 0.0242271 | ko00981 |
| Hepatitis C | 0.0014316 | 0.0245517 | ko05160 |

## 4. Discussion

*L. vannamei* is the most-commonly cultivated shrimp species in China. Following the expansion of the aquaculture scale and the increasing demand for high-quality broodstocks, it is particularly crucial to understand the molecular mechanism of gonadal development and sexual differentiation. MiRNA, with regulatory function, has been widely reported as controlling the gonadal differentiation on aquatic animals. Herein, we conducted a miRNA study to identify differentially expressed miRNAs in gonadal tissue of *L. vannamei* by high-throughput sequencing, and predict the target genes of differentially expressed miRNAs. We also sought to understand the sexual differentiation and development regulatory mechanisms.

Accumulating evidence has suggested that miRNAs play a vital part in regulating target genes expression related to reproductive development. The present study indicated that the miR-9b-3p, miR-1-3p, let-7, and miR-184 families were abundantly expressed in the gonads of the *L. vannamei*. All of these miRNA families were expressed in thousands of fragments. Herein, the miR-184 family was highly expressed in the gonads of mouse [35], *H. cumingii* [18], and swimming crab (*Portunus trituberculatus*) [36]; the miRNA family was also highly expressed in the ovaries of *L. vannamei*. This result was consistent with previous research, which strongly indicated that it may play a vital role in reproductive development. The let-7 family was abundantly expressed in the gonads of blunt snout brean (*Megalobrama amblycephala*) and *A. Latus* [19,37], in accordance with the expression level of miRNAs let-7 identified in gonads of *L. vannamei*. These miRNAs similarly functioned in different species, which revealed the key role of miRNA in reproductive physiology.

We performed a differentially expressed miRNA analysis on the gonads of *L. vannamei*. The miRNAs miR-252 _3, miR-92b-3p_3, miR-133-3p_5, miR-750_1, miR-87a-3p_1, and miR-12a-5p were abundantly expressed in testes. In contrast, the miRNA-252a targeted with cdc2 kinase was involved in the ovarian maturation of Chinese mitten crab (*Eriocheir sinensis*), and regulated the expression of this gene [38]. The miR-92b was highly expressed in the testes of amphioxus (*Branchiostoma japonicum*) [39]. However, miR-92b-3p was significantly up-regulated at the ovarian primordial follicle assembly stage, which indicated that it plays an important role in the reproduction of female mice [40]. Previous studies have confirmed that the miR-133 family plays an important role in reproduction. In human testes, the miR-133b promoted spermatogenesis by targeting downregulation of the gene GLI3 [41]; and miR-133b controlled the expression level of *tagln2* in tilapia, which was further involved in the early oogenesis [42]. In *E. sinensis*, miR-133 exhibited high expression in meiotic maturation of oocytes and played an important role in regulating the 3′-UTR of cyclin B gene [43]. In addition, miR-12 can control the process of ovary activation in worker bees [44]. The findings in these studies were not entirely consistent with our results in *L. vannamei*, suggesting that different miRNAs in a miRNA family may have distinct functions. Meanwhile, the miRNAs miR-750_1 and miR-87a-3p_1 were not found to function in gonads of other species.

In the ovaries of *L. vannamei*, miR-263a-5p_1, miR-279_1, miR-6489-3p, miR-281-2-5p and let-7-5p_6 were highly expressed. The miR-263b were highly expressed in Mud Crab (*Scylla paramamosain*), which negatively regulated the expression of the ERK pathway genes to control ovarian development [45]. Aponetic (APT), a feedback inhibitor, is involved in the JAK/STAT signaling pathway in Drosophila ovary. APT interacted with its downstream target mir-279, thereby limiting JAK/STAT signaling activation, to avoid STAT activity, which disrupts follicle cell identity and cell motility [46]. In *C. hongkongensis*, miR-263b and miR-279 both were abundantly expressed in the ovaries [18]. In *P. monodon*, miR-6489-3p controlled the regulation of *RAs* in the MAPK/ERK pathway, and thus played an essential role in generating oocyte maturation [24]. In *B. japonicum*, miR-281 was regarded as a sexual dimorphism miRNA and showed a higher expression in testes [39], which was contrary to our results in *L. vannamei*. The miR-281 may be involved in the regulation of reproduction and sex determination. The let-7 family was first discovered in the miRNAs group and appeared to control differentiation and development [47]. The miRNA family was highly expressed in the ovaries and testes of *M. amblycephala*, *T. ovatus* and *C. hongkongensis* [20,21,37]. In conclusion, different miRNAs may have functions in regulating ovarian development.

In this study, the miRNAs let-7-5p_6, novel_mir67, miR-12-5p_2, miR-279_1, and miR-92b-3p_3 were highly expressed at stage I ovaries of *L. vannamei*, and novel_mir23 has high expression level at stage I testes, which demonstrated that theses miRNAs may participate in the regulation of early gonadal development. The miR-263a-5p_1 was highly expressed at stage I testes and stage III ovaries, which suggested that it played an important role in early ovarian development and testes maturation.

In total, we predicted 143,319 target genes of differentially expressed miRNAs. These genes mainly enriched the KEGG pathways involved in gonadal differentiation and development, including ECM–receptor interaction, Hedgehog signaling pathway, protein digestion and absorption, and CAMs. In chicken, the ECM–receptor interaction is associated with regulation of the ovary and involved in the egg-laying process [48]. This pathway is also related to sex differentiation in male zebrafish [49], ovarian development and spawn in female scallop *Chlamys farreri* [50]. Hedgehog signaling pathway plays a vital role in the Drosophila testis niche [51]; additionally, this pathway is concerned with sexually dimorphic development of the reproductive organs on mammals [52], including mouse [53] and human [54]. The pathway of protein digestion and absorption enriched by target genes is involved in the ovarian development of *P. trituberculatus* [55] and goats [56], as well as in spermatonesis-regulation in male rats [57]. The CAMs pathway is related to the sex reversal through epigenetic modification in Nile tilapia (*Oreochromis niloticus*) [58]. Our

results revealed out that these target genes significantly enriched in pathways upon gonadal differentiation and development. This finding was consistent with previous studies.

## 5. Conclusions

We identified a total of 463 miRNAs in the ovaries and testes of *L. vannamei* using high-throughput sequencing. Of these miRNAs, 172 miRNAs were significantly differentially expressed between the ovaries and testes. The miR-252-3, miR-92b-3p_3, miR-133-3p_5, miR-750_1, miR-87a-3p_1 and miR-12a-5p were abundantly expressed in testes. The miR-263a-5p_1, miR-279-1, miR-281-2-5p, miR-6489-3p and let-7-5p_6 were highly expressed in ovaries. Furthermore, the target genes of differentially expressed miRNA were enriched in several KEGG pathways involved in gonadal differentiation and development. This finding improves our understanding of the regulation mechanism of gonadal differentiation in the *L. annamei*.

**Supplementary Materials:** The following supporting information can be downloaded at: https://www.mdpi.com/article/10.3390/fishes7060308/s1, Table S1. All differentially expressed miRNAs between ovaries and testes.

**Author Contributions:** Conceptualization, methodology: W.L. and P.H., formal analysis: L.Z., Y.C. and Y.Z., investigation: X.L., C.Y., Q.H. and L.L., resources: B.Z., funding acquisition: P.W., writing—original draft preparation: W.L. and P.H., writing—review and editing: P.W. and J.P., visualization: X.Z. and J.G. All authors have read and agreed to the published version of the manuscript.

**Funding:** This research was funded by the Natural Science Foundation of Guangxi Zhuang Autonomous Region (2019GXNSFBA185022), and the earmarked fund for CARS-48.

**Institutional Review Board Statement:** All procedures involving the handling and treatment of shrimp used in this study were conducted with the approval of the Animal Care and Use Committee of the Guangxi Academy of Fishery Sciences (approval code GXFS-2021-16), Nanning, China.

**Informed Consent Statement:** Informed consent was obtained from all subjects involved in the study.

**Data Availability Statement:** The data presented in this study are openly available in NCBI under bioproject number PRJNA877296.

**Conflicts of Interest:** The authors declare no conflict of interest.

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
