# Peer review of "Identification and Characterization of microRNAs in the Gonads of Litopenaeus vannamei Using High-Throughput Sequencing"

_fishes, doi:10.3390/fishes7060308_

Round 1

Reviewer 1 Report

The authors have completed a study in which they used high-throughput sequencing techniques to identify differently expressed microRNAs in the gonads of male and female shrimp. Seven genes were further analysed using qRT-PCR. I have provided several comments which I hope will help to improve the quality of the manuscript.

Introduction

Please indicate that Litopenaeus vannamei is a species of shrimp in the first sentence. For example, the authors could write: a tropical shrimp species

There are numerous typos and grammatical mistakes throughout the manuscript that should be corrected. I have provided examples below but there are many, many more;

L41: ‘researches’ should be ‘research’

L43: ‘[8, 9]’ should be ‘[8, 9]’

L45: ‘genetic’ should be ‘genetics’

L48: ‘understanding for regulatory’ should be ‘understanding of regulatory’

L48-49: ‘mechanisms of the sexual’ should be ‘mechanisms driving sexual’

L49-50: ‘miRNAs involved in the gonads’ does not make sense and should be re-worded

L47-48: The authors state that ‘gonads develop by regulation of many genes that are differentially expressed between female and male gonads.’ Whilst this is true, there are many other non-gonadally expressed genes that are involved in the development and differentiation of the gonads and this should get a brief mention.

L61-63: The authors provide a list a several scientific species names with no common names. Please either provide the common names for each of these species or indicate whether they are all fish, marine invertebrates, mammals etc. The reader should not be expected to look up all of the individual species names in order to understand how inclusive the statement is.

Methods          

Lots of information is missing from the Methods section. Please provide the following:

Sample sizes (mentioned in Results section but should be mentioned in Methods section)

Thickness of paraffin sections

The model of PCR machine used

PCR cycling parameters

Gel percentages

Names of kits used and the companies that they were purchased from

Indicate where the transcriptome reference sequence can be found – is this available online?

What statistical test was done – SPSS was used, but what test was done?

Table 1 is not mentioned anywhere in the body of text.

Table 1 and Table 2 would be better placed in the Methods section rather than the Results section.

Results

L122-123: The authors state that both the females and males were in ‘stage III’  – please provide a reference explaining all of the different stages of development in both male and female L. vannamei.

If the shrimp used for qRT-PCR were in different stages of development to those used for high-throughput sequencing (stage I vs stage III) then are the results really comparable?

Section 3.1 – the authors should refer to Figure 1 throughout this section so the reader can easily recognise the different cell types being referred to.

Figure 1: The labels on the sections are very hard to see. Please modify the figure so that they are more clear.

L139 refers to Figure 1A – Figure 1A is a micrograph of a gonadal section. Please update to refer to the correct Figure.

The quality of Figure 2 is not good enough – please redo this figure and submit a high quality image where the text is readable.

L161 refers to Figure 1B,C – Figure 1B,C are micrographs of gonadal sections. Please update to refer to the correct Figure.

Section 3.4 – presumably the authors conducted some form of ANOVA on the qRT-PCR results. The F value, degrees of freedom, p value etc. should be reported in the Results section for each significant result.

The quality of Figure 3 is not good enough – please redo this figure and submit a high quality image where the text is readable. As stage I is less developed that stage III it would make sense to present stage I to the left of the figure and stage III to the right of the figure – please reorganise the order of the bars.

All of the figure captions are too brief – especially for Figure 3, there is no mention of species name or what has been plotted (i.e., are the bars showing means +/- standard error? – what is the sample size? – what do the letters above the bars represent?)

The quality of Figure 3 is not good enough – please redo this figure and submit a high quality image where the text is readable.

Discussion

Similar to in the Introduction of the manuscript, the authors present several scientific species names without any mention of the common names leaving the reader to look them up themselves to determine how relevant the species is. Please provide the common names.

Reviewer 2 Report

In this manuscript, the authors identified putative miRNAs with important roles in gonadal development of the shrimp Litopenaeus vannamei. To this end, they use high-throughput sequencing to identify the sex-related microRNAs followed by bioinformatic analyses to elucidate the regulatory mechanisms in which they might be involved during gonadal differentiation and then perform RT-qPCR validations. I found the data interesting and relevant for future research on this organism with commercial interest. It would highly increase the quality of the article to include in situ hybridization for some of the new miRNAs suggested being important for gonadal development. In addition, while the text is very clear and nicely written, the figures present some problems that I state below, together with our comments.

Specific comments:

• Abstract

It would be nice to include the common name (shrimp) in the abstract before the scientific name.

• Keywords

I found it peculiar that the authors refer to "Solexa sequencing" in the keywords when in reality this term will only be referred to in the bibliographic references. In contrast, they often refer to “high-throughput sequencing” and therefore they should probably refer to that in the keywords. In addition, they should include in the methods that they use the Solexa Sequencing.

• Figures

Figure 1

-  A and B panels should be “brighter”.

-  A, B, C, and D lettering should be in the top left corners of each panel.

-  Some numbers in the figure are hard to see in black. One idea is to replace them with the first initials of each word and use a brighter color (white).

Figure 2

-  This figure lost the resolution.

-  Panel A: Authors should say what is “O” and “T” in the legend of the figure or simply use the entire word instead of an abbreviation.

-  Panel B: I think there is a mistake in “up” and “down”, do the authors mean “ovary”, and “testis”?

-  Panel C, very weak resolution, with small lettering.

-  Legend should be improved

Figure 3

-  Lettering is too small

-  The authors here use RT-PCR but in the abstract they use RT-qPCR. They use only the later designation throughout the manuscript

Figure 4

-  Resolution can be improved.

Round 2

Reviewer 1 Report

In my opinion, the authors have significantly improved the quality of their manuscript and it is nearing a point where it could be accepted for publication. There are some minor points that I think could still be improved upon which I have outlined below.

In response to one of my comments on a previous version of this manuscript the authors state that they have included sample size in the Methods section. I cannot see this addition. The number of each sex of shrimp dissected should be included in section 2.1 L80-81, the number of RNA samples from each sex that were mixed together to generate the pooled sample should be included in section 2.2 L93. 

There is still no mention in the methods section of what kind of statistical tests were carried out, whether the assumptions of the test were meet etc. All that is mentioned is that SPSS was used.

I fear the authors have misunderstood my previous request to add statistical outputs into the Results section. To include a Table (i.e., Table 3) reporting the full output is a little unusual (and in this instance, unnecessary). Normal statistical reporting convention dictates that when an ANOVA has been carried out, the F value, degrees of freedom, p value etc. are reported within the Results text for each significant result. i.e., let-7-5p_6 expression was significantly higher in stage I ovaries than in stage I testis (F3,11 = 275.5, p < 0.001). 
